# Antibiotic-Resistant Bacteria in Environmental Water Sources from Southern Chile: A Potential Threat to Human Health

Matías Jofré Bartholin [1], Boris Barrera Vega [2,3] and Liliana Berrocal Silva [1,*]

1   Centro de Investigación en Biomedicina (CIBMED), Escuela de Medicina, Facultad de Medicina, Universidad Finis Terrae, Santiago 7501015, Chile; mjofre@uft.cl
2   Escuela de Tecnología Médica, Facultad de Salud, Universidad Santo Tomás, Santiago 7501015, Chile; bobarrera@hcuch.cl
3   Laboratorio de Microbiología, Hospital Clínico de la Universidad de Chile, Santiago 7501015, Chile
*   Correspondence: lberrocal@uft.cl

**Abstract:** Antimicrobial resistance (AMR) is a critical global issue affecting public and animal health. The overuse of antibiotics in human health, animal production, agriculture, and aquaculture has led to the selection of antibiotic-resistant strains, particularly in Gram-negative bacteria. Mutations and horizontal gene transfer play a significant role in the development of antimicrobial resistance, leading to the reduced efficacy of current antibiotics. Today, AMR in bacteria and antibiotic-resistance genes (ARGs) are increasingly recognized in multiple environmental sources, including recreational and irrigation waters. This study aims to identify Gram-negative bacteria from surface aquatic reservoirs in southern Chile and assess their susceptibility to clinically relevant antibiotics. Water samples were collected from four lakes, five rivers, one waterfall, and one watershed in southern Chile to isolate environmental Gram-negative bacilli (GNB). API-20E and MALDI–TOF were employed for bacterial identification. Kirby–Bauer disc diffusion tests and multiplex PCR were performed to determine their susceptibility profile. A total of 26 GNB strains were isolated from environmental water samples, predominantly belonging to the *Pseudomonas* ($n = 9$) and *Acinetobacter* ($n = 7$) genera. Among these strains, 96.2% were resistant to ampicillin and cefazoline, while 26.9% and 34.6% showed resistance to ceftazidime and cefepime, respectively. Additionally, 38.5% exhibited resistance to colistin. Two *Enterobacter cloacae* strains obtained from Cachapoal River (sixth region) and Villarrica Lake (ninth region), respectively, presented a multidrug-resistant (MDR) phenotype and carried at least two extended-spectrum β-lactamase (ESBL) genes. Thus, antibiotic-resistant GNB and ARGs were found in natural water reservoirs, raising concerns about the dissemination of resistance determinants among potentially pathogenic bacteria in environmental microbial communities.

**Keywords:** aquatic environments; Gram-negative bacilli; antimicrobial resistance; antibiotic-resistance genes; clinically relevant antibiotics; horizontal gene transfer





## 1. Introduction

The emergence and spread of antimicrobial resistance (AMR) in bacteria poses a significant global threat to medical care, with the possibility of a future crisis where no antimicrobial drugs will be effective against multidrug-resistant (MDR) strains. Although both Gram-positive and Gram-negative bacteria contribute to AMR, the prevalence of Gram-negative bacilli (GNB), particularly *Enterobacterales* and non-fermenting bacteria, with resistance to antibiotics like β-lactams and carbapenems, has been increasingly observed in medical practice. These MDR strains are responsible for a subset of pathologies predominantly associated with hospital-acquired infections (HAIs) [1,2]. Critically important pathogens, such as extended-spectrum β-lactamase (ESBL)-producing and Carbapenem-resistant *Enterobacterales*, demand the urgent development of novel therapeutic strategies [3].

The application of selective pressures, like administering antibiotics at defined concentrations, results in the death of a high percentage of sensitive bacteria. However, some

bacteria thrive under these conditions by developing or acquiring resistance strategies encoded by genetic mechanisms, typically located in plasmids and other mobile genetic elements or within the chromosome [4,5]. Infections caused by multiresistant GNB have become an increasingly serious problem both in hospitals and the community. Up to 50% of human-pathogen bacteria may now be resistant to antibiotics commonly used in clinical practice, leading to a three-fold increase in mortality and the risk of complications for affected patients [6].

The lack of new antimicrobial drugs has resulted in the reintroduction of old antibiotics, such as colistin, as a last-resort therapy [7]. However, the continuous discovery of colistin-resistant genotypes among GNB strains in Chile and around the world [8–10] is raising concerns. Preserving the clinical efficacy of these last-resort antibiotics is crucial, given the limited therapeutic alternatives against MDR bacteria.

Growing evidence suggests that AMR may find its origin in the use of antibiotics like β-lactams, polymyxins, and others for livestock. These antibiotics are widely used in animal husbandry and farming, not only for their antimicrobial action against GNB but also for their growth-promoting properties [11–13].

Furthermore, environmental water sources serve as significant reservoirs and transmission vehicles for pathogenic bacteria and antibiotic-resistance genes (ARGs). Effluent from contaminated water treatment plants, containing residues like resistant bacterial cells, antibiotics, disinfectants, and ARGs are discharged into rivers, lakes, and other natural water sources [14,15]. Analyzing environmental waters has proven effective in studying pathogens with antibiotic resistance [16]. In Chile, a few studies have been conducted in this regard, with one of them reporting the presence of ARGs in bacteria isolated from rivers near water treatment plants and in wild bird feces. These findings highlight the dissemination potential of these waters and the presence of ARGs like β-lactamase genes (*bla*SHV, *bla*CTX-M, *bla*KPC, and *bla*TEM) and other determinants of antibiotic resistance commonly used in human and animal medicine (quinolones, tetracyclines, and sulfonamides, among others) [15].

Internationally, colistin-resistance determinants associated with plasmids (*mcr* genes) have been identified in strains of *Escherichia coli* isolated from animals and subsequently found in multiresistant plasmids in various species of the *Enterobacteriaceae* family and other GNB from human samples, animal samples, and meat food samples for human consumption [9]. However, in Chile, there are limited reports on the contribution of environmental samples to the abundance and dissemination of these genes, underscoring the need to understand the resistance profile of environmental bacteria found in close proximity to humans [15,17]. Both in our country and globally, it is necessary to generate greater evidence that explores the contribution of environmental surface waters to the dissemination of AMR-harboring bacteria and ARGs.

In light of the above, the main aim of this study was to identify GNB in water sources of southern Chilean rivers and lakes with antibiotic-resistant profiles commonly observed in medical practice, and to detect the presence of ARGs associated with these resistance phenotypes which can be transferred to pathogenic bacteria. This research contributes to generating evidence that can guide recommendations on the prudent use of antibiotics and to design effective preventive strategies, addressing this crisis related to the treatment of infectious diseases in all relevant sectors, not solely in human health.

## 2. Materials and Methods

### 2.1. Sample Collection

A total of 21 water samples were systematically collected in sterile 15 mL falcon tubes from four lakes, five rivers, one waterfall, and one watershed, covering the sixth to fourteenth regions located towards the south of the capital of Chile, during the daytime between the 5th and 20th of January 2021 (Figure 1 and Table 1), and then stored at 4 °C. These sampling sites supply water for different purposes including recreation, public use like bathing during the summer season, agriculture, and livestock activities. As shown in

Figure 1 and Table 1, a total of 26 GNB strains were isolated from the 21 water samples initially collected.

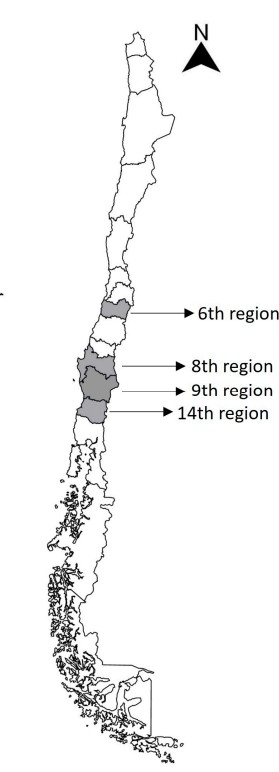

**Figure 1.** Map of Chile. The regions included for sampling are marked in gray, and the sampling locations for each strain obtained are shown on the right.

**Table 1.** Water samples origin, sampling location and selection of strains for identification by MALDI-TOF (*n* = 26).

| N° Correlative Water Sample | Origin of Sample | Sampling Location | Region | Latitude | Longitude | Growth in Blood Agar | Growth in MacConkey Agar | Re-Isolated Strains Name |
|---|---|---|---|---|---|---|---|---|
| 1 | Cachapoal river | Doñihue, Chile | 6th | −34.217.836 | −70.892.976 | yes | yes | 1.4–1.5 |
| 2 | Cachapoal river | Doñihue, Chile | 6th | −34.214.474 | −70.896.846 | yes | yes | 2.1–2.2 |
| 3 | Cachapoal river | Doñihue, Chile | 6th | −34.213.610 | −70.906.275 | yes | yes | 3.1–3.2 |
| 4 | Cachapoal river | Doñihue, Chile | 6th | −34.229.342 | −70.938.954 | yes | yes | 4.2 |
| 5 | Ñuble river | San Nicolás, Chile | 8th | −36.550.236 | −72.094.312 | yes | none | none |
| 6 | Cachapoal river | Doñihue, Chile | 6th | −34.211.686 | −70.873.574 | yes | yes | 6.2 |
| 7 | Cachapoal river | Doñihue, Chile | 6th | −34.209.675 | −70.861.052 | yes | yes | 7.1–7.2 |
| 8 | Bellavista waterfall | Pucón, Chile | 9th | −39.219.463 | −71.844.840 | none | none | none |
| 9 | Coñaripe watershed | Coñaripe, Chile | 14th | −39.582.347 | −72.025.156 | yes | yes | 9.1–9.2 |
| 10 | Calafquen lake | Licán Ray, Chile | 9th | −39.492.032 | −72.161.685 | yes | yes | 10.2 |
| 11 | Voipir river | Villarrica, Chile | 9th | −39.280.841 | −72.306.407 | yes | none | none |
| 12 | Bío-Bío river | San Carlos de Purén, Chile | 8th | −37.599.828 | −72.274.804 | yes | yes | 12.1 |
| 13 | Ñuble river | San Nicolás, Chile | 8th | −36.551.936 | −72.090.913 | yes | yes | 13.1–13.2 |
| 14 | Bío-Bío river | San Carlos de Purén, Chile | 8th | −37.603.116 | −72.268.453 | yes | yes | none |
| 15 | Ñuble river | San Nicolás, Chile | 8th | −36.548.835 | −72.096.363 | yes | yes | 15.1–15.2 |
| 16 | Mulchen-Bureo river | Mulchén, Chile | 8th | −37.719.543 | −72.260.213 | yes | yes | none |
| 17 | Caburgua lake | Caburgua, Chile | 9th | −39.193.653 | −71.796.384 | yes | yes | 17.1–17.2–17.3 |
| 18 | Calafquen lake | Coñaripe, Chile | 14th | −39.565.623 | −72.019.635 | yes | yes | 18.1–18.2 |
| 19 | Villarrica lake | Villarrica, Chile | 9th | −39.275.456 | −72.227.820 | yes | yes | 19.1–19.2–19.3 |
| 20 | Caburgua lagoon | Pucón, Chile | 9th | −39.240.204 | −71.831.649 | yes | none | none |
| 21 | Villarrica lake | Villarrica, Chile | 9th | −39.283.296 | −72.207.432 | yes | none | none |

*2.2. Bacterial Isolation and Preservation*

The collected water samples were cultured on blood agar plates (bioMérieux, Marcy-l'Etoile, France) to promote the recovery of any bacterial species. Subsequently, MacConkey agar plates (Merck, Darmstadt, Germany) were used for the selection and identification of

GNB. The isolated bacterial colonies were characterized based on their ability to ferment lactose, consistency, and ease of growth. The selected bacteria were reisolated in the same medium and then inoculated into 1 mL of LB medium with glycerol (1:1) and stored at −80 °C for further genetic and phenotypic analysis.

### 2.3. Identification of GNB

To achieve bacterial identification, each strain was grown on MacConkey agar plates at 37 °C, and the colonies obtained were used to generate a bacterial suspension (MacFarland 0.5) in sterile saline solution, which was used to sow the API-20E galleries (bioMérieux). In all cases, a numerical code was obtained which was compared in the APIWEB^TM software version number 1.4.1-3 of bioMérieux to achieve the identification of each strain. To corroborate the results obtained through the API galleries, the identification process was carried out using MALDI–TOF (VITEKR MS system, bioMérieux, France). For this, an isolated colony was used on the MacConkey agar plates, along with the *E. coli* ATCC8739 strain, used as a calibrator. This inoculum was allowed to dry, and then, 1 μL of α-cyano-4-hydroxycinnamic acid (HCCA) matrix was added. Subsequently, spectrophotometric readings were taken, and the proteomic spectra were compared using the VITEK® MS Acquisition Station Software version number 1.10-203571250 to facilitate bacterial identification.

### 2.4. Antibiotic Susceptibility Testing

The susceptibility of the identified environmental GNB strains to clinically relevant antibiotics was assessed using agar diffusion assays following the guidelines provided by the Clinical and Laboratory Standards Institute [18]. The antibiotics tested were supplied by Mast Group (Bootle, UK) and included ampicillin (10 μg), cefazoline (30 μg), ceftazidime (30 μg), cefepime (30 μg), imipenem (10 μg), meropenem (10 μg), and ciprofloxacin (5 μg). The susceptibility to colistin was also evaluated using the Sensi-Disc elution method [19]. Bacterial strains resistant to three or more antimicrobial categories were classified as multidrug-resistant (MDR), following standardized criteria [20].

### 2.5. Molecular Detection of Antibiotic-Resistance Genes (ARGs)

Strains exhibiting resistance phenotypes underwent multiplex PCR (mPCR) to detect the presence of different mobile colistin-resistance (*mcr*) alleles and extended-spectrum beta-lactamase (ESBL) genes. The mPCR reactions were carried out using specific primers for detecting the *mcr-1*, *mcr-2*, *mcr-3*, *mcr-4*, and *mcr-5* alleles of the *mcr* genes, as well as the *bla*TEM, *bla*SHV, *bla*CTX-M, *bla*KPC, *bla*VIM, *bla*IMP, *bla*NDM, and *bla*OXA-48 β-lactam resistance genes [21–23]. For this, 1 mL of an overnight bacterial culture was centrifuged, the supernatant was discarded, and the cell pellet was washed with sterile nuclease-free water. Heat shock was then applied to extract the total DNA from the bacteria. Each reaction tube included 12.5 μL of Dream Taq Green PCR Master Mix (Thermo Fisher Scientific, Waltham, MA, USA), 5.5 μL of sterile nuclease-free water, 2 μL of tempering obtained in the previous step, and 0.5 μL of each primer, until obtaining a final volume of 25 μL. Then, the tubes were incubated in a DLAB TC1000s thermocycler, where they were subjected to 1 cycle of denaturation at 94 °C for 15 min followed by 25 cycles of denaturation at 94 °C for 30 s, hybridization at 55 °C for 90 s, elongation at 72 °C for 60 s, and a final elongation cycle of 10 min at 72 °C. The samples obtained from the amplification process were visualized using electrophoresis using 2.5% agarose gels stained with SafeView Plus (FermeloBiotec, Santiago, Chile).

## 3. Results

### 3.1. Distribution and Prevalence of GNB from Environmental Surface Waters

Water samples were systematically collected from various environmental reservoirs, including rivers and lakes, distributed from the sixth to the fourteenth regions of Chile (Figure 1). The sampling areas associated with the sixth region were irrigations and tailing channels near to the Cachapoal River, where strains 1.4 to 7.2 were isolated. Strains

12.1 to 15.2 corresponded to the Bio-Bío and Ñuble rivers in the eighth region. Strains 10.1, 10.2, and 17.1 to 19.3 were obtained from the Caburgua, Calafquén, and Villarrica lakes, respectively, located in the ninth region. Strain 9.1 and 9.2 were obtained from the watershed near to Coñaripe River in the fourteenth region. A total of 26 bacterial strains were isolated from these water samples (Table 1). The identified GNB bacteria mainly belonged to non-fermenting bacilli and the *Enterobacterales* order. The most frequently observed species were *Pseudomonas aeruginosa/fluorescens/putida/mosselli* (34.6%), *Acinetobacter pittii/calcoaceticus/haemolyticus* (27%), and *Stenotrophomonas maltophilia* (11.5%). Strains of *Enterobacter cloacae*, *Rhanella aquatilis*, *Serratia marcescens*, and *Pantoea agglomerans*, representing the *Enterobacterales* order, were also identified (Table 2).

**Table 2.** Identification of GNB obtained from environmental surface waters and antibiotic susceptibility profile (*n* = 26).

| Strain | Identification | Resistance Profile | *BLEE* Genes |
|---|---|---|---|
| 1.4 | *Pseudomonas aeruginosa/fluorescens/putida/mosselli* | AMP–CFZ–FEP–COL | NF |
| 1.5 | *Pseudomonas aeruginosa/fluorescens/putida/mosselli* | AMP–CFZ–FEP–COL | NF |
| 2.1 | *Pseudomonas aeruginosa/fluorescens/putida/mosselli* | AMP–CFZ–COL | NF |
| 2.2 | *Pseudomonas aeruginosa/fluorescens/putida/mosselli* | AMP–CFZ–MEM–COL | NF |
| 3.1 | *Comamonas aquatica* | CIP | NF |
| 3.2 | *Enterobacter cloacae* | AMP–CFZ–CAZ–FEP–IPM–MEM–CIP | *blaCTX–M–blaTEM* |
| 4.2 | *Pseudomonas aeruginosa/fluorescens/putida/mosselli* | AMP–CFZ–COL | NF |
| 6.2 | *Pseudomonas aeruginosa/fluorescens/putida/mosselli* | AMP–CFZ | NF |
| 7.1 | *Acinetobacter pittii/calcoaceticus/haemolyticus* | AMP–CFZ–COL | NF |
| 7.2 | *Acinetobacter pittii/calcoaceticus/haemolyticus* | AMP–CFZ–COL | NF |
| 9.1 | *Acinetobacter pittii/calcoaceticus/haemolyticus* | AMP–CFZ–CAZ | *blaTEM* |
| 9.2 | *Rahnella aquatilis* | AMP–CFZ | NF |
| 10.2 | *Pseudomonas aeruginosa/fluorescens/putida/mosselli* | AMP–CFZ | NF |
| 12.1 | *Stenotrophomonas maltophilia* | AMP–CFZ–CAZ–FEP | NF |
| 13.1 | *Pantoea agglomerans* | AMP–CFZ–FEP | *blaCTX–M* |
| 13.2 | *Pseudomonas aeruginosa/fluorescens/putida/mosselli* | AMP–CFZ | NF |
| 15.1 | *Stenotrophomonas maltophilia* | AMP–CFZ | NF |
| 15.2 | *Pseudomonas aeruginosa/fluorescens/putida/mosselli* | AMP–CFZ | NF |
| 17.1 | *Acinetobacter pittii/calcoaceticus/haemolyticus* | AMP–CFZ–CAZ | NF |
| 17.2 | *Stenotrophomonas maltophilia* | AMP–CFZ–CAZ | NF |
| 17.3 | *Acinetobacter pittii/calcoaceticus/haemolyticus* | AMP–CFZ–FEP–COL | NF |
| 18.1 | *Acinetobacter pittii/calcoaceticus/haemolyticus* | AMP–CFZ–CAZ | NF |
| 18.2 | *Acinetobacter pittii/calcoaceticus/haemolyticus* | AMP–CFZ–MEM–COL | NF |
| 19.1 | *Enterobacter cloacae* | AMP–CFZ–CAZ–FEP–IPM–MEM–CIP | *blaCTX–M–blaTEM* |
| 19.2 | *Rahnella aquatilis* | AMP–CFZ–FEP | NF |
| 19.3 | *Serratia marcescens* | AMP–CFZ–FEP–COL | NF |

Abbreviations: AMP, ampicillin; CFZ, cefazoline; CAZ, ceftazidime; FEP, cefepime; IMP, imipenem; MEM, meropenem; CIP, ciprofloxacin; COL, colistin; and NF, not found.

### 3.2. Resistance Profile of Environmental GNB

To determine the susceptibility profiles of the isolated strains to clinically relevant antibiotics, their resistance to ampicillin, cefazolin, ceftazidime, cefepime, imipenem, meropenem, ciprofloxacin, and colistin was evaluated. The majority of the GNB isolates (96.2%) showed resistance to ampicillin and cefazoline, both β-lactam antibiotics. Resistance to other cephalosporins, ceftazidime and cefepime, was observed in 26.9% and 34.6% of the strains, respectively. A small number of isolates were resistant to carbapenems, with two strains showing resistance to imipenem (7.7%) and four strains showing resistance to meropenem (15.4%). Regarding fluoroquinolones, 11.5% of the isolates exhibited resistance to ciprofloxacin. Notably, 38.5% of the isolates (10/26) showed resistance to colistin (Table 3).

Ruling out the intrinsic resistance observed in most of the environmental isolates (AMP, CFZ), 73.1% of the strains (19/26) were found to be non-susceptible (resistant or intermediate) to at least one antibiotic. Furthermore, we found that two of the studied strains (7.7%) exhibited an MDR phenotype according to standardized criteria [20].

**Table 3.** Susceptibility profile of environmental isolates against clinically relevant antibiotics.

| Resistance Phenotype | % (Frequency) |
|---|---|
| Ampicillin | 96.2% (25/26) |
| Cefazoline | 96.2% (25/26) |
| Ceftazidime | 26.9% (7/26) |
| Cefepime | 34.6% (9/26) |
| Imipenem | 7.7% (2/26) |
| Meropenem | 15.4% (4/26) |
| Ciprofloxacin | 11.5% (3/26) |
| Colistin | 38.5% (10/26) |

The susceptibility profile was obtained using agar diffusion assays. For colistin, Sensi-Disc elution tests were performed.

*3.3. Colistin and β-Lactam Resistance Genes in Environmental GNB*

To investigate certain transferable genetic determinants underlying colistin and β-lactam resistance, the presence of some ARGs was evaluated. Several *mcr* alleles (*mcr-1* to *mcr-5*), responsible for plasmid-mediated mobile colistin resistance, were searched through the use of mPCR. However, none of the evaluated strains were found to carry those alleles. To correlate the resistance phenotypes against ceftazidime and cefepime with the presence of mobile elements in GNB, mPCR was conducted to identify the *bla*TEM, *bla*SHV, and *bla*CTX-M genes, encoding extended-spectrum β-lactamases (ESBLs). Only 15% of the tested isolates (3.2, 9.1, 13.1, and 19.1) were found to carry ESBL genes. The *bla*CTX-M gene was detected in strain 13.1, which corresponds to *Pantoea agglomerans*. Strain 9.1, identified as *Acinetobacter* sp., carried the *bla*TEM gene. Additionally, strains 3.2 and 19.1, both identified as *Enterobacter cloacae*, carried both the *bla*TEM and *bla*CTX-M genes (Table 2). Likewise, mPCR was performed on imipenem- and meropenem-resistant isolates to identify the *bla*KPC, *bla*VIM, *bla*IMP, *bla*NDM, and *bla*OXA-48 genes encoding carbapenemases, but none were found.

**4. Discussion**

The presented descriptive study focused on the prevalence of GNB in environmental waters in Chile and their susceptibility to clinically relevant antibiotics. The bacterial populations in 21 surface water samples from Chilean rivers and lakes were dominated by Gamma-proteobacteria, with *Pseudomonas*, *Acinetobacter*, *Stenotrophomonas*, and some *Enterobacterales* as the main genera. This finding aligns with studies from other countries, where similar bacterial genera have been found in various aquatic environments [24]. However, unlike a previous study conducted by Díaz-Gavidia C. et al., 2021, in Chile [17], we did not find clinically relevant *Enterobacterales* such as *E. coli*, *Citrobacter* sp., and *Klebsiella* sp. These differences could be attributed to the geographic region of sampling and the seasonal variation, since in our study all samples were taken in the summer, in high temperatures and low rain conditions. Whereas, in the study conducted by Díaz-Gavidia C. et al. (2021), the samples were taken during the autumn and winter.

The non-fermenting bacteria *Acinetobacter*, *Pseudomonas*, and *Stenotrophomonas* are opportunistic pathogens found in both humans and animals, capable of acquiring multiple resistance determinants such as β-lactamases or carbapenemases [25–27]. The combination of intrinsic and acquired resistance in these bacteria can lead to therapeutic failure [28,29]. Some species like *Pseudomonas aeruginosa* and *Stenotrophomonas maltophilia* have been reported as MDR and even pandrug-resistant in clinical isolates [27,30,31], and are listed by the World Health Organization as important Gram-negative MDR bacteria that cause healthcare-associated infections [3]. Additionally, in this study, we found some *Enterobacterales* like *Enterobacter cloacae*, *Rhanella aquatilis*, *Pantoea agglomerans*, and *Serratia marcescens*, which can also cause infections in immunocompromised hosts [32–35]. Our finding of *E. cloacae* with an MDR profile in samples obtained from Villarrica Lake (ninth region) and Cachapoal River (sixth region) (Table 2) is concerning, as it can cause both

healthcare-associated and community-acquired infections, such as urinary tract infections and bacteremia [36].

The current study revealed a high percentage of resistance to β-lactam antibiotics like ampicillin and first-generation cephalosporins, which is consistent with natural resistance found in environmental bacterial populations [26,37–39]. Resistance to third- and fourth-generation cephalosporins was also frequent, along with some resistance to carbapenems. Additionally, a portion of the isolates showed resistance to ciprofloxacin and colistin, although no alleles associated with mobile resistance were detected. Therefore, colistin resistance would be attributed to the presence of some intrinsic or acquired resistance mechanisms encoded at the chromosome level in these isolates [7]. Among the evaluated ARGs, our study found ESBL-coding genes (*bla*TEM and *bla*CTX-M) in some strains, indicating potential horizontal gene transfer between environmental and pathogenic bacteria. Consistent with this finding, ESBL-coding genes have been previously found in water sources and non-clinical environments in different countries around the world [1,14,40–42].

In Chile, recent studies have reported a high incidence of *bla*CTX-M in different environments such as in domestic and wild animals [43–45], vegetables, and water [17]. In addition, *bla*TEM has been found in rivers of the fourteenth region [15]. It is interesting to note that we found both *bla*TEM and *bla*CTX-M in *Enterobacter cloacae* and *bla*TEM in *Acinetobacter* sp., two antibiotic-resistant GNB with pathogenic potential and that are associated with healthcare settings. However, we detected *bla*CTX-M in *Pantoea agglomerans*, a frequent saprophytic bacterium found in a wide variety of environments that occasionally causes infections [35,46]. This finding suggests that in aquatic environments, ARGs could be transferred from environmental bacteria to potential pathogenic bacteria and vice versa through horizontal gene transfer mechanisms. Even if such genes have rarely been identified in bacteria of animal and environmental origin, they may sooner or later spread to this reservoir. Some studies indicate that these are promiscuous species in terms of the acquisition or transference of resistance determinants by means of plasmids and other mobile genetic elements [24,47]. Such transmission between environmental and clinically relevant bacteria could have implications for public health and require ongoing surveillance. Considering these findings, this study highlights the importance of a one health approach, recognizing the interconnection between human, animal, and environmental health. The presence of antibiotic-resistant bacteria and ARGs in rivers and lakes that supply water for various purposes, including recreation, agriculture, and public use, may pose risks to human and animal health. It is crucial to monitor and understand the dynamics of antibiotic resistance in environmental microbial communities, as has been suggested previously [1,47–50], to design effective preventive strategies as part of national action plans to address the spread of resistance.

## 5. Conclusions

We observed that 100% of the identified GNB in the water samples presented a resistance phenotype to at least one antibiotic, and in four of them ESBL-coding genes were found. Two of these strains, obtained from Cachapoal River and Villarrica Lake, identified as *E. cloacae*, showed an MDR phenotype and were positive for the *bla*TEM and *bla*CTX-M genes. In conclusion, this indicates that the studied aquatic environments are contaminated with antibiotic-resistant GNB, and the two aforementioned reservoirs are also contaminated with ARGs.

Overall, these findings highlight the need for continued research and proactive measures to address AMR in both clinical and non-clinical settings. Considering the impact of antibiotic-resistant bacteria and ARGs on human, animal, and ecological health, surveillance is crucial to determine contamination levels in aquatic environments. It is a matter of public concern to provide information about the transmission of ARGs among environmental bacteria, whose presence represents a risk to human health. Furthermore, this study reflects the reality of GNB and the distribution of certain ARGs during a specific period of the 2021 summer season in Chile, so it would be important to contrast this evidence

with the current situation during the same summer period. This considers the speed at which bacterial species can evolve in relation to their environmental adaptation. Finally, it should be noted that, although the samples obtained are not representative of all aquatic areas in the country, it would be interesting to increase the sampling of environmental surface waters and the identification of GNB not only in the south but also in the central and northern regions of Chile, in order to correlate the presence of ARG-harboring species with different environmental factors and also relate them to lateral gene transfer events.

**Author Contributions:** Conceptualization, M.J.B. and L.B.S.; methodology, M.J.B., B.B.V. and L.B.S.; investigation, M.J.B., B.B.V. and L.B.S.; writing—original draft preparation, M.J.B. and L.B.S.; writing—review and editing, M.J.B., B.B.V. and L.B.S.; supervision, L.B.S.; project administration, M.J.B. All authors have read and agreed to the published version of the manuscript.

**Funding:** This research was funded by CAI-2020 to M.J.B. and L.B.S., a project that was financed by Universidad Finis Terrae.

**Institutional Review Board Statement:** Not applicable.

**Informed Consent Statement:** Not applicable.

**Data Availability Statement:** Not applicable.

**Acknowledgments:** We would like to thank Alondra Cárdenas and Martín Villavicencio for the sample collection from southern Chile and Arturo Chávez for his valuable comments on the manuscript.

**Conflicts of Interest:** The authors declare no conflict of interest.

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
