# Peer review of "Antibiotic-Resistant Bacteria in Environmental Water Sources from Southern Chile: A Potential Threat to Human Health"

_2036-7481, doi:10.3390/microbiolres14040121_

Round 1
Reviewer 1 Report
Comments and Suggestions for Authors
Journal: Microbiology Research (ISSN 2036-7481)
Manuscript ID: microbiolres-2629248
Type: Research Article
Manuscript title: Antibiotic-Resistant Bacteria in Environmental Water Sources from Southern Chile: A Potential Threat to Human Health
Recommendation: Minor Revision
The work presented deals with an interesting area of microbiology. The manuscript with the title, “Antibiotic-Resistant Bacteria in Environmental Water Sources from Southern Chile: A Potential Threat to Human Health” Studies contains significant scientific findings that have its own kind of importance. Antimicrobial resistance is a definitely a global issue that affect the overall public concern. There are a lot of misuse and overuse of antibiotics that is one of main reason to increase antibiotic-resistance in bacteria.
The title, objectives, hypothesis and experiment design of paper are appropriate.
Paper is very well written; however, some points need due attention which I mention below:
- What you only isolate gram-negative bacteria and test antibiotic resistance against them?
- In Figure 1, You must provide latitude and longitude. You can also make a separate table for this.
- Please construct a phylogenetic tree of isolates using MegaX or any other software.
- Conclusions need to be improved by specifying the discussed important points within the work. In this section, the authors should also provide an outlook on the challenges and potential future directions.
- Although, the paper contains a good number of recent citations and bibliographic assembly.
- A deep review of grammar, punctuation, and spelling is needed across the manuscript.
I also suggest careful proof-checking of the entire manuscript.
Comments on the Quality of English Language
The quality of English language is average, it can be improved in the revision.
Author Response
Thank you very much for taking the time to review this manuscript. Please see the attachment and the corresponding corrections in the re-submitted file.

Reviewer 2 Report
Comments and Suggestions for Authors
This paper investagated the detection of susceptibility to clinically relevant antibiotics of Gram-negative bacteria.The study is useful for evaluate the impact of antibiotic-resistant bacteria on human health. However,the design of the experiment,results and discussion was not well demonstrated in this version of the manuscript. There are also many spelling mistakes.Some issues need to be revised or corrected and several questions need clarification.
1. The introduction section is not well written.Please clarify the reason why this paper focused on the antibitic resistance Gram-negative bacteria, not the Gram-posotive bacteria in the introduction.
2. The main novelty of this study should be clarified in the introduction.
3. p93 and p95,collected in sterile 15 mL falcon tubes,the sentence is repeated twice.
4. How to select the sampling sites? Why only2 samples were colleceted from 14th region?
5. The common water characters, such pH, temperatur, COD, total cell number and the bactrerail communities, et al are needed.
6. Section 2.5, why did the authors choose these ARGs, are they representitives of the ARGs profile in water sample? I do not think these genes could comprehensively delineate the overall situation.
7. Only 26 strains were isolated from 26 water sample, I don’t think the number is enough to describe the antibic resistance bacteria in the water samples.
8. The relationship between the resistance chataters of GNB and the water parameters should be well demostated.
9. The description of the mechanism of resistance in The 3.3 section is insufficient.
10.There is not sufficent information in the conclusion section. Please revise the conclusion to emphasize your main results.
Comments on the Quality of English LanguageModerate editing of English language required
Author Response

(The authors gave the same response as above.)

Reviewer 3 Report
Comments and Suggestions for Authors
The manuscript Antibiotic-Resistant Bacteria in Environmental Water Sources 2
from Southern Chile brings a little information regarding what is happening in the south Chile Waters. The information presented is important, not only to the scientific community but also to the public. Although the manuscript is well written and presented, I leave some questions and suggestion on the PDF

Comments on the Quality of English LanguageGood
Author Response

(The authors gave the same response as above.)

Reviewer 4 Report
Comments and Suggestions for Authors
This study focuses on an antibiotic resistance bacteria in aquatic environments. It is an important topic. However more characteristics of the study area should be added. This can explain the type and amount of bacteria in flowing (rivers) and still (lakes) surface waters or watersheds. Please complete this and detailed comments are provided below.
The methodology should include information on the manufacturer, place and country of production, e.g. ready-made tests used, etc.
Detailed comments:
1. Line 20 – please add from how many lakes and rivers water samples were collected.
2. Line 27 – what was a characteristic of this 6th and 9th regions? The region numbers mean nothing to the readers.
3. Line 31 – “aquatic environments” must be first, please remove “Environmental bacteria”.
4. Line 73 – beta or β
5. Line 94 – please add from how many lakes and rivers water samples were collected. Please give some characteristic of these regions. How did the regions differ? Why were these regions selected for research? Was there a difference in the analysis between the type and amount of bacteria in flowing surface water (rivers) and standing water (lakes) or watershed?
6. Line 95 – where is figure 1 and table 1? They should be right below the reference to them.
7. Figure 1 - Next to Figure 1, please enlarge the 4 regions examined and mark the differences between them. If possible, please provide geographical coordinates.
8. Line 318 – in how many water samples?
9. Line 325 - what were the seasonal variations? How did the temperature and precipitation change? Were the detected bacteria observed in high/low temperature conditions or during heavy rains etc.?
10. Please include in your conclusions what corrective measures the authors suggest to use and whether they plan to expand research on antibiotic resistant bacteria in environmental water sources in the future.
Author Response

(The authors gave the same response as above.)

Round 2
Reviewer 2 Report
Comments and Suggestions for Authors
In this new version, most suggestions have been revised. However, the results didn’t show enough important conclusions. Deep analysis based on the results is insufficiency,especially the mechanism of resistance in the 3.3 section.
Comments on the Quality of English LanguageIn this new version, most suggestions have been revised. However, the results didn’t show enough important conclusions. Deep analysis based on the results is insufficiency,especially the mechanism of resistance in the 3.3 section.
Author Response

(The authors gave the same response as above.)

Reviewer 4 Report
Comments and Suggestions for Authors
I propose to accept the manuscript in present form.
Author Response

(The authors gave the same response as above.)
